# M3Exam: A Multilingual, Multimodal, Multilevel Benchmark for Examining Large Language Models

**Wenxuan Zhang**[1], **Sharifah Mahani Aljunied**[1], **Chang Gao**[1,2*], **Yew Ken Chia**[1,3*], **Lidong Bing**[1]

[1]DAMO Academy, Alibaba Group
[2]The Chinese University of Hong Kong   [3]Singapore University of Technology and Design
{saike.zwx, mahani.aljunied, gaochang.gao, yewken.chia, l.bing}@alibaba-inc.com

## Abstract

Despite the existence of various benchmarks for evaluating natural language processing models, we argue that human exams are a more suitable means of evaluating general intelligence for large language models (LLMs), as they inherently demand a much wider range of abilities such as language understanding, domain knowledge, and problem-solving skills. To this end, we introduce M3Exam, a novel benchmark sourced from real and official human exam questions for evaluating LLMs in a multilingual, multimodal, and multilevel context. M3Exam exhibits three unique characteristics: (1) multilingualism, encompassing questions from multiple countries that require strong multilingual proficiency and cultural knowledge; (2) multimodality, accounting for the multimodal nature of many exam questions to test the model's multimodal understanding capability; and (3) multilevel structure, featuring exams from three critical educational periods to comprehensively assess a model's proficiency at different levels. In total, M3Exam contains 12,317 questions in 9 diverse languages with three educational levels, where about 23% of the questions require processing images for successful solving. We assess the performance of top-performing LLMs on M3Exam and find that current models, including GPT-4, still struggle with multilingual text, particularly in low-resource and non-Latin script languages. Multimodal LLMs also perform poorly with complex multimodal questions. We believe that M3Exam can be a valuable resource for comprehensively evaluating LLMs by examining their multilingual and multimodal abilities and tracking their development. Data and evaluation code is available at https://github.com/DAMO-NLP-SG/M3Exam.

## 1 Introduction

In recent years, large language models (LLMs) have demonstrated remarkable capabilities in various natural language processing (NLP) tasks [9, 31, 3, 4]. For instance, ChatGPT [30] shows an impressive ability to effectively respond to a wide range of questions and provide high-quality answers [7]. Their applications even extend beyond traditional NLP domains, as they have been integrated to address real-world challenges in diverse areas [22, 27, 40]. Given the increasing reliance on LLMs, the need for appropriate and comprehensive evaluations has become ever more critical. Such assessments should not only examine whether the models exhibit strong language understanding, but also evaluate their capabilities to handle complex problems requiring different kinds of skills [31].

Typically, NLP models are evaluated using well-designed benchmarks for specific tasks, such as SQuAD [34] for question-answering or WMT [8] for machine translation. Although useful, these task-specific benchmarks often emphasize on certain aspects, and thus do not adequately assess the breadth

---

*Chang Gao is a research intern at Alibaba. Yew Ken Chia is under the Joint Ph.D. Program between Alibaba and SUTD.

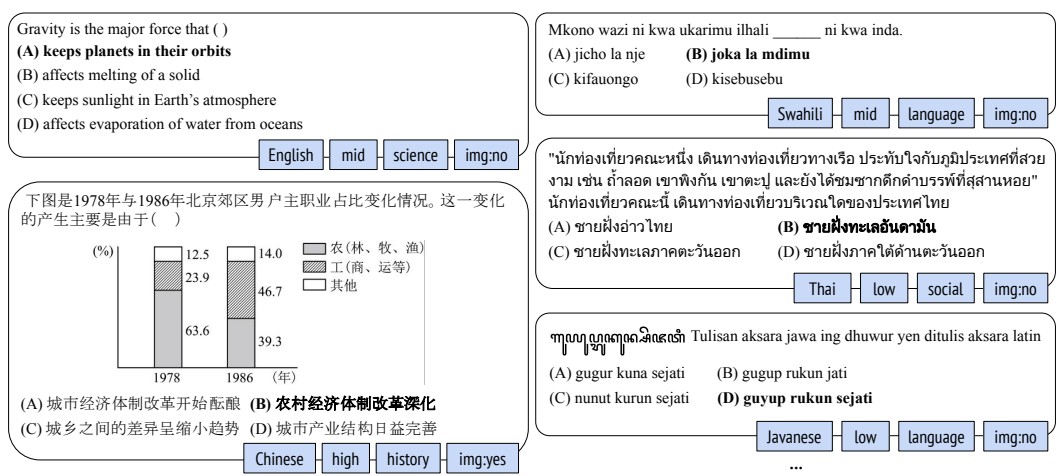

Figure 1: Example questions from M3Exam dataset. Correct answers are indicated in bold. Meta-information is provided with blue boxes attached to each question.

of abilities possessed by powerful modern LLMs. In comparison, tackling human exam questions often require diverse skills such as language understanding, complex reasoning, etc. Therefore, a new trend has emerged to utilize tests originally designed for humans to assess the performance of LLMs. For example, MMLU [17] contains exam questions covering 57 tasks across a diverse set of subjects for assessing models. AGIEval [43] collects questions from standardized exams such as law school admission tests and math competitions. GPT-4 [31] also uses a variety of human exams to test its ability in complex scenarios. These human-centric evaluations, approximating real-world applications and expectations, have been demonstrated to be valuable testbeds for gauging the artificial general intelligence (AGI) capabilities of LLMs.

Despite the advantages of evaluations based on human exams, current benchmarks exhibit several key limitations. Firstly, the majority of these benchmarks focus on questions in English [17, 43], neglecting the evaluation of a model's performance in a multilingual context. As many LLMs exhibit multilingual ability and are widely used across different countries and languages [30, 3, 4], it is important to evaluate their multilingual capabilities. Moreover, some existing multilingual benchmarks of traditional NLP tasks [14, 33, 31] that were created by translating original English datasets have been found to introduce an English-centric bias. This bias arises because the translation process, while making the benchmarks available in multiple languages, does not always capture culturally specific or unique concepts present in the target languages. This shows the importance of sourcing real data from various languages to represent their native cultural background [28]. Secondly, most benchmarks consider solely text-based questions, ignoring a significant portion of real-world exam questions include images. Considering this type of question is essential to test a model's multimodal understanding abilities in a wide range of practical applications [15]. Lastly, existing exam-type benchmarks usually draw from mixed exams such as college final exams or professional certificate tests [17, 43], the constructed resources are thus also comprised of questions from mixed levels. Gathering exam questions from varying educational levels is critical to assess and understand the level of intelligence that LLMs have developed. [10].

In this paper, we present M3Exam, a novel benchmark dataset designed for evaluating the artificial general intelligence of large language models. M3Exam has several unique characteristics: (1) Multilingualism - by gathering questions from official exams across multiple countries, the benchmark contains natural multilingual questions in different languages, which retain the social-cultural diversity of knowledge that may be essential for problem-solving; (2) Multimodality - we incorporate all types of questions including those that require images and carefully process these images to facilitate convenient model evaluation. We show that a significant proportion (approximately 23%) of questions demand information from images for solving; (3) Multilevel structure - we adopt a top-down approach for data collection where we first select three critical educational periods (primary, middle, and high school) and source official exams from the culmination of each period, resulting in a benchmark with varying levels. In total, M3Exam comprises 12,317 questions in 9 diverse languages, with 2,816 questions involving one or more images. Each question includes the question text, candidate answer

options, ground-truth answer, and rich meta-information consisting of language, education level, subject, and whether images are involved. Some examples are shown in Figure 1.

We utilize a wide range of top-performing LLMs in both multilingual and multimodal settings to assess their performance on the newly introduced M3Exam dataset. Our findings indicate that the majority of existing models have difficulties in processing multilingual text, with GPT-4 [31] being the only model to achieve over 60% accuracy. Nevertheless, it still faces challenges with low-resource languages such as Javanese, and non-Latin script languages like Thai. Current multimodal models also underperform on M3Exam, with state-of-the-art models such as BLIP-2 [26] attaining less than 50% accuracy. A detailed examination further reveals that comprehending complex images and reasoning across images remain quite challenging for current models. Moreover, we surprisingly find that LLMs' performances do not show a monotonic decrease with the educational level increases which is quite different from human behavior, implying that the development of intelligence in LLMs may not necessarily align with that of human intelligence. Overall, we believe M3Exam can serve as a valuable resource for examining LLMs, both tracking their improvements in terms of multilingual and multimodal settings and providing insights into the development of model intelligence with different education levels.

## 2  M3Exam Benchmark Dataset

### 2.1  Design Principle

Exams are widely used to assess human intelligence at various educational stages, as they draw on the integration of diverse skills, including language understanding, world knowledge, cultural awareness, and logical reasoning, etc. Consequently, exam questions offer an ideal testbed for evaluating the general intelligence of LLMs. We propose three crucial design principles for constructing the M3Exam benchmark dataset with the exam questions:

- **Multilingual Evaluation**: While most existing datasets primarily focus on English, assessing LLMs' abilities in multiple languages with different cultural backgrounds, especially those low-resource languages, is crucial to apply LLMs in broad-range scenarios. To achieve this, collecting real-world natural data of different languages instead of translating from English data is of great importance, as culture and world knowledge are deeply rooted in authentic data [28].

- **Multimodal Evaluation**: In real-world scenarios, humans often encounter problems with different modalities such as images or audio. Multimodal evaluation is thus essential for testing an LLM's ability to jointly process information from multiple modalities, which reflects a key part of cognition capability. Therefore, to facilitate such evaluation, we include questions requiring images for successful solving.

- **Multilevel Evaluation**: Although education systems vary across countries, they typically organize learning into several stages (e.g., from primary school to middle school and then to high school), with examinations to assess students' readiness to advance to the next stage. The exams at the end of each period effectively reveal the general intelligence expectations within each country. Thus, evaluating LLMs with questions from these critical educational stages offers a comprehensive assessment of their capacity with different levels of intelligence requirements.

### 2.2  Language Selection and Data Collection

Following the design principle outlined above, we adopt a top-down approach to construct our dataset. To comprehensively evaluate the model, we select 9 languages including English (from the US), Chinese (from China), Italian (from Italy), Portuguese (from Brazil), Vietnamese (from Vietnam), Thai (from Thailand), Swahili (from Kenya), Afrikaans (from South Africa), and Javanese (from Indonesia). This selection is mainly driven by language and cultural diversity, with the aim of covering different language families, languages with varying levels of resources, written scripts, and their major spoken countries.

We then engage native speakers from each of the selected countries to collect official exam papers along with their answers at the end of each educational level, which are typically the graduation exams of primary school, middle school, and high school. We encourage them to 1) choose exams with the largest possible participation (e.g., if a period has two exams, one nationwide and one statewide, the

Table 1: Data statistics of M3Exam dataset. We rank languages by their ratio in the CommonCrawl corpus ("CC Size"), and report the detailed number of questions at each level: X/Y denotes there are X questions involving only pure text, and Y questions requiring images to solve.

| Language | Code | Country | CC Size | Low | Mid | High | Total |
|---|---|---|---|---|---|---|---|
| English | en | US | 46.175 | 306 / 106 | 505 / 204 | 1132 / 485 | 1943 / 795 |
| Chinese | zh | China | 4.632 | 83 / 14 | 347 / 335 | 281 / 104 | 711 / 453 |
| Italian | it | Italy | 2.726 | 220 / 107 | 291 / 140 | 318 / 160 | 829 / 407 |
| Portuguese | pt | Brazil | 1.131 | 86 / 96 | 182 / 98 | 645 / 278 | 913 / 472 |
| Vietnamese | vi | Vietnam | 1.056 | 170 / 16 | 361 / 12 | 1286 / 88 | 1817 / 116 |
| Thai | th | Thailand | 0.440 | 472 / 113 | 568 / 174 | 1154 / 114 | 2194 / 401 |
| Swahili | sw | Kenya | 0.008 | 186 / 4 | 248 / 0 | - | 434 / 4 |
| Afrikaans | af | South Africa | 0.007 | 91 / 36 | 138 / 63 | 54 / 64 | 283 / 163 |
| Javanese | jv | Indonesia | 0.004 | 205 / 4 | 172 / 1 | - | 377 / 5 |
| Total | | | | 1819 / 496 | 2812 / 1027 | 4870 / 1293 | **9501 / 2816** |

nationwide exam should be collected); 2) collect all available subjects and up to five papers across different years for each subject to ensure a diverse range of questions. In the end, we collect in total 435 exam papers from nine countries. The details of those exam papers are in Appendix A.4.

## 2.3  Data Processing and Annotation

Given the diverse languages we consider, many collected exam papers are only available as images or in scanned versions. Therefore, we first conduct OCR to convert these papers into editable text versions. The original papers, editable text versions, and corresponding answer sets are then passed on to annotators of each specific language, who transform the data into a unified format. In terms of question scope, we focus on multiple-choice questions, as they allow for a standard automatic evaluation of the correctness of model outputs. We exclude subjective questions with free-response answers but include questions that can be easily adapted into the multiple-choice format, such as judging true-false statements.

Specifically, the annotators are asked to check the text content and fix potential errors due to OCR transformations. Then they need to separate the question text from a list of candidate answer options, and input the correct answer. We also address a limitation observed in previous benchmarks: inadequate or limited context information. Many questions require rich contextual information to answer, such as reading comprehension questions with passages or chemistry problems featuring brief introductions to new chemical phenomena. We specifically ask annotators to include such contextual background information. Furthermore, we also convert special formats into pure text, such as converting all equations into LaTeX format or using 
 to include a text span for representing a bold font. All these format adaptations aim to make the constructed benchmark mimic the real exam scenario. Multiple rounds of quality checks are conducted to ensure the data quality.

To accommodate questions containing both text and images, we instruct annotators to annotate images with placeholders, regardless of whether the images appear in the question text or option text. This ensures clarity on whether an image is needed and its original placement in the question. For example, (image)[image-x.jpg] will appear in the place of an image in the transformed text, and the corresponding image will be clipped and saved with the same name (i.e., image-x.jpg).

## 2.4  Data Statistics

At the end of the annotation and quality check, our newly introduced M3Exam dataset contains a total of 12,317 questions. Each question includes context information, the main question text, candidate options, the correct answer, and meta information, such as its language, level, subject, and whether images are needed to solve the question. Figure 1 shows examples of some questions.

Table 1 provides detailed statistics of M3Exam, broken down by language and level. The number of questions that involve only pure text, or require images are separately listed. We rank languages by their ratio in the CommonCrawl corpus, which is a widely-used data source for training LLMs. It can be observed that our selected languages span a wide range, from high-resource languages like English and Chinese, to extremely low-resource languages such as Javanese. Therefore, the diversity

of selected languages makes it well-suited for comprehensively assessing the multilingual capabilities of the model. The ratio of questions requiring images also varies across countries, from over 60% questions with images for Chinese, to languages with very few image-type questions. After obtaining the data, we group the questions for each language into four subject categories, namely language, math, social science, and natural science. We then randomly select three questions for each subject category of each level in each language and separate these as held-out development data, which can be used as in-context examples. The remaining questions are used as test data during the experiment.

# 3 Experiment Setups

## 3.1 Models

To evaluate the performance of various LLMs on our newly introduced M3Exam dataset, we select a range of top-performing models in either multilingual or multimodal settings.

**Text-only LLMs**   To process multilingual texts, we first take ChatGPT (`gpt-3.5-turbo`) [30] and GPT-4 (`gpt-4`) [31] from OpenAI, both of which have demonstrated strong multilingual abilities in preliminary studies [2, 7, 24, 31]. Additionally, we also adopt Claude (`Claude-instant`) from Anthropic, a model which is often considered to be comparable to ChatGPT [19]. We obtain the results of those close-source models via API call with the corresponding model type. Furthermore, we utilize two open-source models, namely BLOOM (`176B`) [36] and Vicuna (`13B`) [12]. BLOOM stands out as one of the largest open-source LLMs specializing in multilingual ability, having been trained with data encompassing 46 languages and 13 programming languages. Vicuna, on the other hand, was developed by fine-tuning the LLaMA model [37] on user-shared conversations. Although not specifically designed as a multilingual model, recent leaderboards have identified Vicuna as the top-performing open-source model on both English-only and non-English leaderboards [29].

**Multimodal LLMs**   To evaluate LLMs on multimodal questions, we consider a range of state-of-the-art open-source models since closed-source models such as GPT-4 do not have official multimodal versions available currently. Specifically, we employ BLIP-2 [26] and InstructBLIP [15], which have demonstrated leading performance in various multimodal question-answering tasks. However, these models are limited to processing a single image per question. Since our M3Exam data may contain multiple images in the background description or as answer options, we additionally utilize Fromage [23] and OpenFlamingo [6], both of which are capable of handling multi-image inputs. We use their pre-trained model weights to directly conduct inference on our test data, and further impose a constraint decoding to generate only valid multiple-choice options for those models.

## 3.2 Settings

**Zero-shot Evaluation**   We primarily evaluate various LLMs in zero-shot settings. There are three considerations for this setup decision: First, using a zero-shot approach to prompt the model mimics the natural process in real-world applications and the problem-solving process in exams. Secondly, most LLMs have limited context lengths (especially for multilingual models handling diverse languages) or cannot accept multiple images as input (for multimodal models), rendering them unsuitable for evaluation using multiple few-shot demonstrations. Third, since the majority of existing LLMs have undergone instruction tuning [32], they are readily capable of following instructions to output in the desired format. Nonetheless, we also compare zero-shot and few-shot settings from an empirical standpoint with ChatGPT in Section 4.1.

**Prompt**   Following the convention of previous studies [17, 31], we clearly specify the subject type of each question by starting with "*The following is a multiple choice question about {subject type}.*". Subsequently, we include an instruction

The following is a multiple choice question about Social Science. Please only give the correct option, without any other details or explanations.

What is a 100-year period of time called?
(A) decade      (B) millennium
(C) century      (D) light-year

Answer: C

- - - - - - - - - - - - - - - - - - - - - - - - - - - -

以下是关于数学的单项选择题，请仅给出正确选项对应的选项序号而非其他细节。

将3克药放入100克水中，药与药水的比是
(A) 3:97  (B) 3:100  (C) 3:103

答案:C

Figure 2: Illustrations of the prompt with two short questions. Model outputs are marked in green. Each option will take a new line in practice.

Table 2: Results on questions of different languages. Accuracy scores are reported.

| | en | zh | it | pt | vi | th | sw | af | jv | avg |
|---|---|---|---|---|---|---|---|---|---|---|
| random | 25.01 | 25.93 | 33.77 | 21.41 | 25.21 | 22.89 | 25.00 | 25.05 | 25.00 | 25.47 |
| passing | 60.00 | 60.00 | 60.00 | 60.00 | 50.00 | 50.00 | 40.00 | 50.00 | 60.00 | 54.44 |
| BLOOM | 28.62 | 29.47 | 33.17 | 7.20 | 23.81 | 9.09 | 27.10 | 23.26 | 26.95 | 23.19 |
| Vicuna | 56.99 | 29.18 | 35.39 | 41.73 | 27.33 | 15.08 | 24.07 | 33.33 | 27.49 | 32.29 |
| Claude | 74.25 | 51.61 | 61.90 | 62.54 | 51.65 | 31.27 | 38.32 | 63.95 | 30.73 | 51.80 |
| ChatGPT | 75.98 | 61.00 | 67.94 | 62.43 | 57.18 | 34.09 | 53.04 | 68.99 | 37.47 | 57.57 |
| GPT-4 | 87.55 | 79.47 | 83.23 | 74.24 | 70.49 | 56.04 | 65.89 | 84.11 | 55.26 | **72.92** |

"*Please only give the correct option, without any other details or explanations.*" to constrain the model output for automatic evaluations. A question is then presented, along with its corresponding options each in a new line. Finally, the prompt ends with "*Answer:*" for the model to generate its output. It is important to note that all prompts are language-specific [31]. We translate the prompt for each language to ensure that the entire prompt presented to the model is monolingual. This prompt design is the same in both multilingual and multimodal settings, except we omit the format constraint for multimodal experiments as constraint decoding is applied. Two example prompts are shown in Figure 2. Detailed prompts as well as examples of different prompting strategies are provided in Appendix A.2.

**Evaluations** As all the questions are multiple-choice questions, we utilize accuracy as the evaluation metric. In most cases, the models can adhere to the instructions and produce only the option. Consequently, we take the first alphabetic letter of the model's output as the prediction and compare it with the ground truth answer to calculate the accuracy scores.

## 4 Results and Discussions

### 4.1 Multilingual Evaluation

**Main multilingual results** We present the results of various LLMs on different language data in Table 2. We also show the scores of random guesses ("random") and the conventional scores that are considered as passing the exam ("passing").[2] Overall, we observe that most models can only achieve less than 60% accuracy, with GPT-4 being a notable exception, achieving 72.92% and consistently outperforming all other models across different languages. BLOOM, although a multilingual model, gives unsatisfactory performance and is even worse than the random guess since it may generate invalid options. ChatGPT, Claude, and Vicuna show varying degrees of performance depending on the language. Vicuna, despite having a much smaller model size, gives reasonable performance for Latin-script languages. While ChatGPT and Claude have relatively similar performance in English (75.98% v.s. 74.25%), ChatGPT demonstrates better results in other languages, suggesting a more robust multilingual ability. When comparing performance across different languages, we observe that existing models generally perform worse for non-Latin languages, such as Chinese (despite being relatively high-resource), as well as low-resource languages like Javanese (even though it mostly uses the Latin script). In summary, the results on our newly introduced M3Exam dataset highlight the challenges and limitations faced by current LLMs in handling non-Latin and low-resource languages, suggesting that there is still a large room for improvement in their multilingual capabilities.

**Handling non-English questions with different prompting strategies** In multilingual settings, some pilot studies have discovered that using English task instructions [24] or employing a translate-test approach (i.e., translating target language data to English) [2] can lead to improved performance compared to using monolingual prompts in a specific language. To analyze the impact of different prompting strategies, we follow such two settings to create another two types of prompts for ChatGPT, denoted as "EN-Instruct" and "EN-Translation"[3], respectively. Detailed examples of these two types

---

[2]Note that the exact passing line depends on each specific exam. Here we provide the scores conventionally used in the corresponding countries, which can indicate the relative difficulty of the questions.

[3]We use Google Translation API (`https://translate.google.com/`) for translating the data into English.

Table 3: Results on different prompting strategies based on ChatGPT.

| Prompt | en | zh | it | pt | vi | th | sw | af | jv |
|---|---|---|---|---|---|---|---|---|---|
| Monolingual | 75.98 | 61.00 | 67.94 | 62.43 | 57.18 | 34.09 | 53.04 | 68.99 | 37.47 |
| EN-Instruct | - | 60.56 | 69.30 | 61.42 | 57.57 | 32.70 | 49.30 | 70.16 | 38.27 |
| EN-Translation | - | 57.92 | 62.76 | 59.62 | 56.40 | 46.49 | 48.13 | 70.16 | 50.94 |
| Few-shot | 75.46 | 60.26 | 64.36 | 62.99 | 58.64 | 37.41 | 51.87 | 67.05 | 33.42 |

of prompts are given in Figure 6 in the Appendix, and the results are presented in Table 3, where we also show the performance of the original prompt ("Monolingual"). We can note that using English instructions does not consistently improve performance, potentially because our data originates from actual language data rather than merely translated English data. Consequently, using English prompts may not better elicit the knowledge required to solve the questions. The impact of using translated data ("EN-Translation") varies across different languages. On one hand, many questions are closely tied to each specific language, translated data may lose essential information in such cases, leading to poorer performance. On the other hand, translations could eliminate some barriers to understanding particular languages, especially those that the ChatGPT model struggles with, such as Thai and Javanese. Therefore, using the English translations of the questions greatly improves their performance.

**Zero-shot v.s. few-shot setting** To empirically investigate the impact of few-shot demonstrations, we run experiments on both zero-shot and few-shot settings with ChatGPT. Specifically, we use the held-out few-shot samples for each language, and append the few-shot samples after the instruction but before the final testing sample (see Figure 6 for detailed examples). The format for the few-shot samples is the same as the final test sample, except that the correct option is given after "Answer:" for those samples. We present the results of few-shot samples in Table 3, denoted as "Few-shot". It can be noticed that introducing few-shot examples does not necessarily lead to performance improvement on average. While for languages such as Portuguese and Vietnamese, prompting with few-shot examples result in an improvement, the model's performance in other languages such as Chinese and Swahili slightly decreases with few-shot demonstrations. The reason might be that existing LLMs are already familiar with the question format of human exams. Thus using in-context demonstrations does not provide any additional advantages. Moreover, the effectiveness of few-shot learning depends on many factors such as language complexity, the model's knowledge, the selection of few-shot examples, etc.

## 4.2 Multimodal Evaluation

Table 4: Results on questions with images. We report the performance on both questions with a single image ("Single"), multiple images ("Multi"), as well as the overall scores ("Overall").

| Model | Size | # Input img | Single | Multi | Overall |
|---|---|---|---|---|---|
| random | - | - | 25.00 | 25.00 | 25.00 |
| Flan-T5 | 11B | 0 | 49.70 | **40.34** | 48.30 |
| ChatGPT | NA | 0 | 60.36 | 28.57 | 55.60 |
| Fromage | 7B | many | 21.45 | 30.25 | 22.77 |
| OpenFlamingo | 9B | many | 30.18 | 27.73 | 29.81 |
| BLIP-2 | 12B | 1 | **51.18** | 36.97 | **49.06** |
| InstructBLIP | 12B | 1 | 48.82 | 32.77 | 46.62 |

In Table 4, we present the performance of various models on English questions, as there are no existing LLMs handling both multilingual and multimodal settings. In addition to multimodal models, we provide random guess baselines, the performance of the Flan-T5 model (XXL version) [13], and the performance of ChatGPT. Although Flan-T5 is a text-only model, it has a similar parameter size to the selected multimodal models and serves as the text encoder for both BLIP-2 and InstructBLIP, making it a suitable comparison baseline. Similarly, we also use ChatGPT to understand the extent to which it can perform using only text-based inputs. We only feed the text part for each question to these two text-only models. For BLIP-2 and InstructBLIP models, we only take the first image as the input as they can only process a single image.

The following are multiple-choice questions about natural-science.

The diagram represents a light ray reflecting from a plane mirror. The angle of reflection for the light ray is ( )

**(A) 25°**  (B) 35°
(C) 50°  (D) 65°

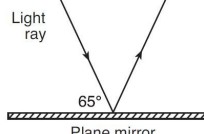

Light ray

65°

Plane mirror

Model Outputs:

Fromage: D  /  Flamingo: B
BLIP-2: D  /  InstructBLIP: D

*What does the image show? Describe all details including the figures, texts and numbers.*

Fromage: Here's what it's supposed to look like: Lines under the mirrors are.

Flamingo: Draw a horizontal line so it will fall on the right side of the small object. Draw a line so that it will fall on the other side of the image. Together, draw a perpendicular line from the top of the image from the cross line between the two lines formed with the last two steps.

BLIP-2: a diagram showing the light ray path and path length through a plane mirror

Instruct-BLIP: plane mirror

Figure 3: An example question with different model outputs (in green) on the left-hand side, as well as model outputs when asked to describe the image in detail on the right-hand side.

We observe that most models do not yield satisfactory performance in general. When compared to Flan-T5, only the BLIP-2 model marginally surpasses its performance. This outcome is unexpected, as Flan-T5 can only process text as input and ignore the images, which intuitively suggests that it may lose crucial information. Upon closer examination, we discover that all existing multimodal models struggle to comprehend complex image details in exam questions (e.g., axis details in math questions, map details in geography questions), which are vital for various subjects. We present an example question and the corresponding outputs from different models in Figure 3. To further assess the extent to which models understand the image used in this question, we construct a new prompt: "*What does the image show? Describe all details, including figures, texts, and numbers. Answer:*" to gauge the models' behavior. As demonstrated in the right portion of the figure, only BLIP-2 captures relatively more accurate information about the image. However, none of the models can accurately discern details such as the marked angle 65°, making it impossible for them to solve this question.

For questions involving multiple images, the difficulty increases as cross-image reasoning becomes necessary. However, Fromage and OpenFlamingo, models specifically designed for handling multiple images, do not demonstrate clear improvements. Instead, they perform notably worse than BLIP-2 and InstructBLIP, which are only capable of handling single images. We find that they often struggle to comprehend even individual image details (as shown in the example in Figure 3). This finding suggests that pre-training on multiple images does not necessarily guarantee better multimodal understanding abilities. Overall, in comparison to existing multimodal datasets consisting of relatively simple visual question-answering tasks [5, 16], our M3Exam dataset presents a significant challenge to understanding image details and reasoning under cross-image and cross-modal settings. We provide more examples and discussions in Appendix A.3.

## 4.3 Multilevel Evaluation

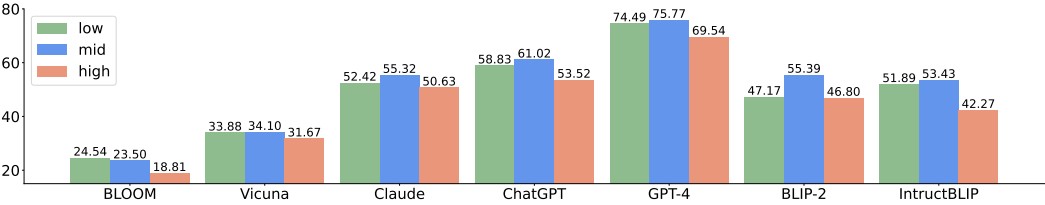

Figure 4: Performance of different LLMs broken down along different levels.

One advantage of the M3Exam dataset is that it encompasses questions from three critical educational periods, namely low, mid, and high, which represent varying levels of difficulty. We here examine LLMs on questions from these three levels. The results are summarized in Figure 4. Comparing the performance on three different levels, the high level indeed generally has the lowest performance, showing its difficulty. Surprisingly, for almost all LLMs, whether text-only or multimodal models, there is no clear decreasing trend as the level increases. This observation contrasts with conventional human behaviors. For example, a high-school student who can achieve reasonable scores in graduation

exams should achieve much better results in exams of lower-level schools. Consequently, we expect that human performance will exhibit a monotonic decrease as the level increases.

This result suggests that although LLMs show impressive results on many tasks and are even said to spark artificial general intelligence [10], the emergence and development of intelligence in LLMs have significant differences from that of human intelligence and require further investigations. This is also reasonable since the "learning process" of LLMs is different from humans. They are typically trained on massive data first, making their knowledge heavily biased towards the data that are more common, while humans often learn from easy principles and knowledge to more complex reasoning and thinking skills. Moreover, this finding indicates that creating more challenging datasets might not be efficient for improving the models [42, 25, 21]. Instead, it might be more crucial to investigate the underlying reasons for LLM failures, even at primary school-level questions, and devise strategies to address these shortcomings.

### 4.4 Discussions

**Performance across various subjects** In an effort to better understand the proficiency of models across different subject types, we evaluated ChatGPT's performance in four languages with diverse levels of resources, including English (en), Chinese (zh), Vietnamese (vi), and Thai (th). The results, as displayed in Figure 5, reveal some intriguing patterns. Notably, across all languages, the model tends to underperform in the math category. This suggests that the reasoning skills required in these questions present a great challenge for the model. Conversely, the model exhibits relatively stronger performance in the natural science and social science subjects across all languages, indicating a more effective handling of structured and factual information in these areas.

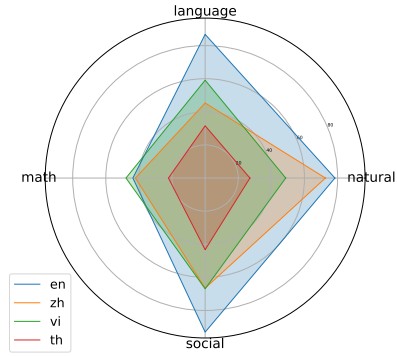

Figure 5: Performance of ChatGPT across different subject categories.

## 5 Related Work

Large language models (LLMs) have witnessed remarkable advancements in recent years, enabling them to generate human-like text, answer complex questions, and perform a wide range of NLP tasks. These models, such as GPT-3 [9], Claude [4], GPT-4 [31], and PaLM2 [3] have demonstrated exceptional performance on various benchmarks and have been widely adopted in academia and industry. However, the evaluation of these models is a critical aspect that requires careful consideration to ensure reliable and comprehensive assessments.

For the evaluation of NLP models, traditional approaches primarily rely on established NLP benchmark datasets. Popular benchmarks such as GLUE [39], SuperGLUE [38], and SQuAD [34] focus on specific NLP tasks, such as question answering, sentiment analysis, and text classification. To facilitate multilingual evaluation, researchers have also developed multilingual benchmarks such as XTREME [18] and XTREME-R [35]. These benchmarks provide standardized evaluation settings, diverse language coverage, and task-specific evaluation metrics to assess models' performance in a multilingual setting [2, 7, 24]. In the multimodal context, the evaluation often involves assessing the model's ability to understand and generate content that combines multiple modalities, such as text, images, and videos. Some typical evaluation tasks include image captioning [11, 1], image question answering [5, 16], visual reasoning [20], video question answering [41] etc.

Although performance on typical benchmark datasets provides valuable insights into the capabilities of LLMs, it may not be sufficient to evaluate their general intelligence in real-world scenarios. To bridge this gap, there has been a growing trend of utilizing exams originally designed for humans to evaluate LLMs in recent times. An early work is the MMLU [17] dataset, which collects questions covering 57 tasks to test the model's world knowledge and multitask accuracy. More recently, similar benchmark datasets have been proposed following this direction, such as AGIEval [43] with various types of exams, C-Eval [19] and GAOKAO [42] benchmarks using exam questions in Chinese to evaluate Chinese LLMs, and IgakuQA [22] that evaluates ChatGPT on Japanese Medical Licensing Exams. However, these datasets suffer from several limitations, including limited language diversity,

the absence of multimodal evaluation, and the lack of multi-level evaluation. These limitations restrict the comprehensive assessment of LLMs in real-world scenarios.

## 6 Conclusions

We introduce M3Exam in this work, a novel benchmark dataset for evaluating LLMs by offering a multilingual, multimodal, and multi-level assessment. Our analysis of top-performing LLMs on M3Exam reveals that current models face challenges in processing multilingual text, especially in low-resource and non-Latin script languages. Additionally, state-of-the-art multimodal models struggle to achieve reasonable accuracy on M3Exam. Overall, it provides a valuable resource for tracking the progress of LLMs in multilingual and multimodal settings and offers insights into the development of model intelligence across various education levels. However, M3Exam only considers multiple-choice questions for now, making it unsuitable to evaluate LLMs for questions requiring creative writing. We will consider such questions in our future work.

## Acknowledgements

We extend our gratitude to the data management and annotation team at Alibaba DAMO Academy, particularly Yanyan Zheng, Tantong Champaiboon, Nguyen Ngoc Yen Nhi, and Andila Putri Susanti. Their kind assistance in coordinating the annotators, processing the data, and conducting quality checks has been important to our work. We would also like to thank the annotators from various linguistic and cultural backgrounds who collaborated to contribute to this unique multilingual dataset.

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

# A   Appendix

## A.1   Dataset Documentation

We provide additional information on the introduced M3Exam dataset in this section.

### A.1.1   Motivation

M3Exam is created to test models' multilingual and multimodal abilities through questions that are from real and official human exams. Current benchmarks on multilingual and/or multimodal evaluation still mainly focus on traditional NLP tasks, which have several limitations as described in the paper, we aim to bridge this gap through M3Exam. Moreover, we aim to provide insights into the development of machine intelligence through questions from different education levels.

### A.1.2   Composition

- M3Exam contains textual questions, part of them need images to solve.
- There are 12,317 questions in total.
- Questions are from exam papers across multiple years for each language, thus they are representative of the expected knowledge of certain languages.
- M3Exam is self-contained. Part of the questions requiring images are released with the corresponding images and clearly identified.
- The dataset does not involve any specific person and does not contain any information that might be offensive, insulting, or threatening.

### A.1.3   Usage and Distribution

- The dataset is released at `https://github.com/DAMO-NLP-SG/M3Exam`.
- The data is saved in JSON format, where an example is shown in the README.md file. An example code snippet is also provided showing how to read and process the data.
- License: M3Exam is under CC BY-NC-SA License.

## A.2   Examples of different prompting strategies

We show some detailed examples of different prompting strategies in Figure 6. It can be noticed from the example of the Thai question, that even when translating the data into English, the question might still be difficult to answer since it may involve background knowledge of each specific language.

## A.3   More examples on multimodal questions

We present two additional examples of questions involving images in Figure 7. In the first question, even though an image is required to answer, the keyword "Gandhi" is already mentioned in the question text. As a result, a text-only model like Flan-T5 might be capable of providing the correct answer. Upon further examination of the models' descriptions, we observe that BLIP-2 and Instruct-BLIP offer relatively more accurate descriptions. BLIP-2 refers to nonviolent resistance or civil resistance, demonstrating its ability to capture the semantic or conceptual meaning of the images. Instruct-BLIP, on the other hand, provides a detailed description of the physical activity depicted in the figure. However, neither Fromage nor Flamingo delivers relevant descriptions of the photograph, or gives hallucination descriptions.

In the second example, we select a question with images present in the options. To obtain the descriptions, we separately feed each image to the model and show models' generations of two option images for simplicity. It is worth noting that for this relatively easy image understanding task, all models seem to provide more accurate descriptions. They can recognize the physical objects in the images and even identify the specific number of cubes in some cases. However, it is still difficult for them to consistently give accurate descriptions for all options. Overall, we can see that multimodal questions in M3Exam post a great challenge for existing multimodal models compared to previous multimodal tasks since they require a more accurate understanding of the involved images and may even need to reason across multiple images.

| Zero-Shot Prompting | Few-Shot Prompting | English-Instruction |
|---|---|---|

**Zero-Shot Prompting**

The following is a multiple choice question about Social Science. Please only give the correct option, without any other details or explanations.

What is a 100-year period of time called?
(A) decade
(B) millennium
(C) century
(D) light-year

Answer:

- - - - - - - - - - - - - - - - - - - - - - - - -

以下是关于数学的单项选择题，请仅给出正确选项对应的选项序号而非其他细节。

将3克药放入100克水中，药与药水的比是
(A) 3:97
(B) 3:100
(C) 3:103

答案：

- - - - - - - - - - - - - - - - - - - - - - - - -

ต่อไปนี้เป็นคำถามแบบปรนัย วิชาสังคมศึกษา. โปรดระบุคำตอบเป็นตัวเลือกที่ถูกต้องโดยไม่ต้องให้รายละเอียดอื่นเพิ่มเติม.

"นักท่องเที่ยวคณะหนึ่ง เดินทางท่องเที่ยวทางเรือ
ประทับใจกับภูมิประเทศที่สวยงาม เช่น ถ้ำลอด เขาพิงกัน
เขาตะปู และยังได้ชมซากตึกดำบรรพ์ที่สุสานหอย"
นักท่องเที่ยวคณะนี้ เดินทางท่องเที่ยวบริเวณใดของ
ประเทศไทย
(A) ชายฝั่งอ่าวไทย
(B) ชายฝั่งทะเลอันดามัน
(C) ชายฝั่งทะเลภาคตะวันออก
(D) ชายฝั่งภาคใต้ด้านตะวันออก

คำตอบ:

**Few-Shot Prompting**

The following are multiple choice questions about Social Science. Please only give the correct option, without any other details or explanations.

Which term is used to describe money collected to pay for the services that a community provides?
(A) savings
(B) profit
(C) interest
(D) taxes

Answer: D

John's country has a prime minister and a Parliament that plan, organize, and make decisions. What does this statement describe about John's country?
(A) its interdependence
(B) its geography
(C) its population
(D) its government

Answer: D

What is a 100-year period of time called?
(A) decade
(B) millennium
(C) century
(D) light-year

Answer:

**English-Instruction**

The following is a multiple choice question about Math. Please only give the correct option, without any other details or explanations.

将3克药放入100克水中，药与药水的比是
(A) 3:97
(B) 3:100
(C) 3:103

Answer:

**English-Translation**

The following is a multiple choice question about Math. Please only give the correct option, without any other details or explanations.

Put 3 grams of medicine into 100 grams of water, the ratio of medicine to medicine water is
(A) 3:97
(B) 3:100
(C) 3:103

Answer:

- - - - - - - - - - - - - - - - - - - - - - - - -

The following is a multiple choice question about Social Science. Please only give the correct option, without any other details or explanations.

"A group of tourists travels by boat, admiring the beautiful landscape, such as Tham Lot Cave, Khao Phing Kan, Khao Ta-Pu, and also visiting the fossils at the Shell Cemetery." In which area of Thailand do these tourists travel?
(A) Gulf of Thailand coast
(B) Andaman Sea coast
(C) Eastern Seaboard coast
(D) Eastern coast of Southern Thailand

Answer:

Figure 6: Detailed examples of different prompting strategies. "Zero-shot Prompting" refers to our default prompting strategy, where prompts are in each specific language. "Few-shot Prompting" involves few-shot in-context examples, which appear before the final test sample. "English-instruction" uses English instructions instead of instructions in the target language. "English-Translation" uses the translated English data of other languages.

## A.4 Details on exam papers

We list the specific exams we collected for constructing the datasets in Table 5.

## A.5 Examples questions of each language in M3Exam

We list example questions of each language from Figure 8 to Figure 16.

The following are multiple-choice questions about social-science.

Mahatma Gandhi Spinning Yarn in the Late 1920s

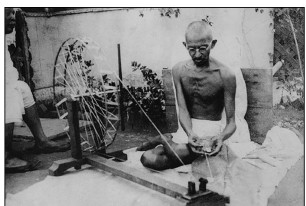

*Source: Carnegie Council for Ethics in International Affairs, January 15, 1929"*

This photograph is most often associated with the term

(A) appeasement     (B) containment
(C) boycott         (D) segregation

Model Outputs:

Fromage: D  /  Flamingo: D
BLIP-2: C  /  InstructBLIP: C

*What does the image show? Describe all details including the figures, texts and numbers.*

Fromage: This is a textile, and not a photograph.

Flamingo: The picture shows famous Bharat Ratna(India\u2019s premiere civilian award) awardee, Rashtrapati Dr. A.P.J. Abdul Kalam, better known as Missile Man of India, preparing tea for his students!! How is this a question? No description is given. No mention is made of details such as the figures, texts, and numbers are showing. The important details in this question are only the figures and texts, and a hint concerning the nature of the numbers. The figures and texts refer to the details of the image: Dr Kalam is holding a tea

BLIP-2: hand spinner who is seen wearing saffron robes was satyagrahis

Instruct-BLIP: man spinning wheel

---

The following are multiple-choice questions about math.

In her math class, Carla used unit cubes to build a right rectangular prism with a volume of 24 cubic units. The height of the prism was two units. Which figure could be the bottom layer of the prism?

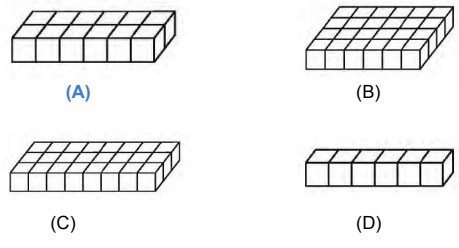

(A)        (B)

(C)        (D)

Model Outputs:

Fromage: A  /  Flamingo: B
BLIP-2: A  /  InstructBLIP: A

*What does the image show? Describe all details including the figures, texts and numbers.*

Fromage:
- (A) The structure has equal squares which are all arranged according to their mental angle.
- (C) The illustration indicates the general arrangements of a rectangle.

Flamingo:
- (A) We can see a 12 cube. Then there are three figures making up the cube in various positions. One cube is facing up. One cube is facing right. And one cube is facing down.
- (B) This CD shows a period of four years. The four years are represented by stages. Each stage is represented by a box

BLIP-2:
- (A) a drawing of white square blocks each one square
- (B) The image shows an illustration of a 4 x 4 cube with a line coming from one side going over to the other

Instruct-BLIP:
- (A)-(D) cubes

Figure 7: Two examples of questions involving images. We show the original questions, the model outputs, as well as the model responses when asked to describe the images in detail.

---

Gravity is the major force that ( )

(A) keeps planets in their orbits

(B) affects melting of a solid

(C) keeps sunlight in Earth's atmosphere

(D) affects evaporation of water from oceans

English   mid   science   img:no

Figure 8: An example question in English, with the correct answer in bold and blue.

Table 5: List of exams (translated into English) for constructing the M3Exam dataset.

| Language | Level | Exam |
|---|---|---|
| English | low | New York State Testing Program Grade 5 |
|  | mid | New York State Testing Program Grade 8 |
|  | high | High School Regents Examinations |
| Chinese | low | Beijing Admission Examination for Junior High Schools (grade 6) |
|  | mid | Beijing Senior High School Entrance Examination (grade 9) |
|  | high | National College Entrance Examination (NCEE) (grade 12) |
| Italian | low | State exam of Primary School (grade 5) |
|  | mid | State exam of First Level Secondary School (grade 8) |
|  | high | State exam of Second Level Secondary School (grade 10) |
| Portuguese | low | Primary School Exams (grade 5 and 6) |
|  | mid | National Youth and Adult Competency Certification Exam (Encceja), Primary Education (grade 9) |
|  | high | National High School Exam (ENEM) |
| Vietnamese | low | Primary School Semester II Final Exam (grade 5) |
|  | mid | Secondary Graduation Exam, Secondary School Semester II Final Exam (grade 9) |
|  | high | National High School Graduation Exam (grade 12) |
| Thai | low | O-NET (Ordinary National Educational Test) for Primary 6 (grade 6) |
|  | mid | O-NET (Ordinary National Educational Test) for Secondary 3 (grade 9) |
|  | high | O-NET (Ordinary National Educational Test) for Secondary 6 (grade 12) |
| Swahili | low | Kenya National Examinations Board Exam (KNEB), Kenya Primary School Education Assessment (KPSE) (grade 6) |
|  | mid | Kenya Certificate of Primary Education (KCPE) (grade 8) |
| Afrikaans | low | Provincial Examinations, Intermediate Phase (grade 6) |
|  | mid | Provincial Examinations, Senior Phase (grade 9) |
|  | high | National Senior Certificate (grade 12) |
| Javanese | low | End of Semester I/II Exam (UAS), School Final Exam for Elementary School (grade 6) |
|  | mid | End of Semester I/II Exam (UAS), School Final Exam for Junior High School (grade 9) |

下图是1978年与1986年北京郊区男户主职业占比变化情况。这一变化的产生主要是由于 ( )

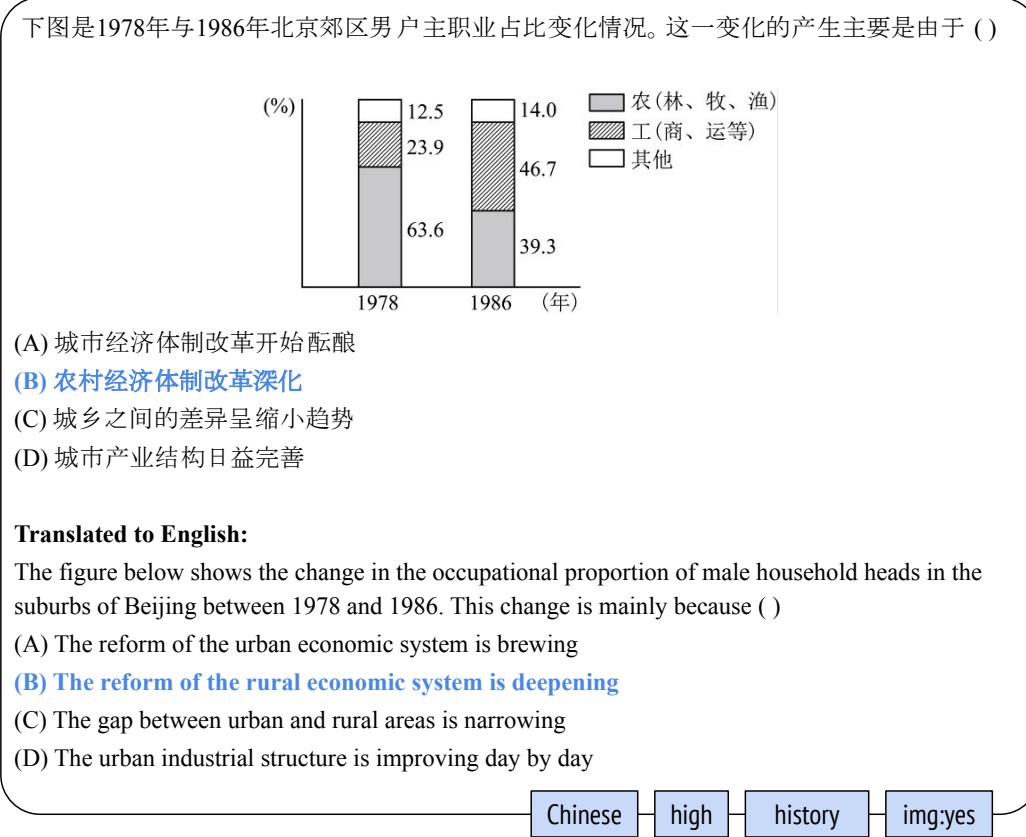

(A) 城市经济体制改革开始酝酿

**(B) 农村经济体制改革深化**

(C) 城乡之间的差异呈缩小趋势

(D) 城市产业结构日益完善

**Translated to English:**

The figure below shows the change in the occupational proportion of male household heads in the suburbs of Beijing between 1978 and 1986. This change is mainly because ( )

(A) The reform of the urban economic system is brewing

**(B) The reform of the rural economic system is deepening**

(C) The gap between urban and rural areas is narrowing

(D) The urban industrial structure is improving day by day

Chinese | high | history | img:yes

Figure 9: An example question in Chinese, with the correct answer in bold and blue.

Osserva la seguente figura.

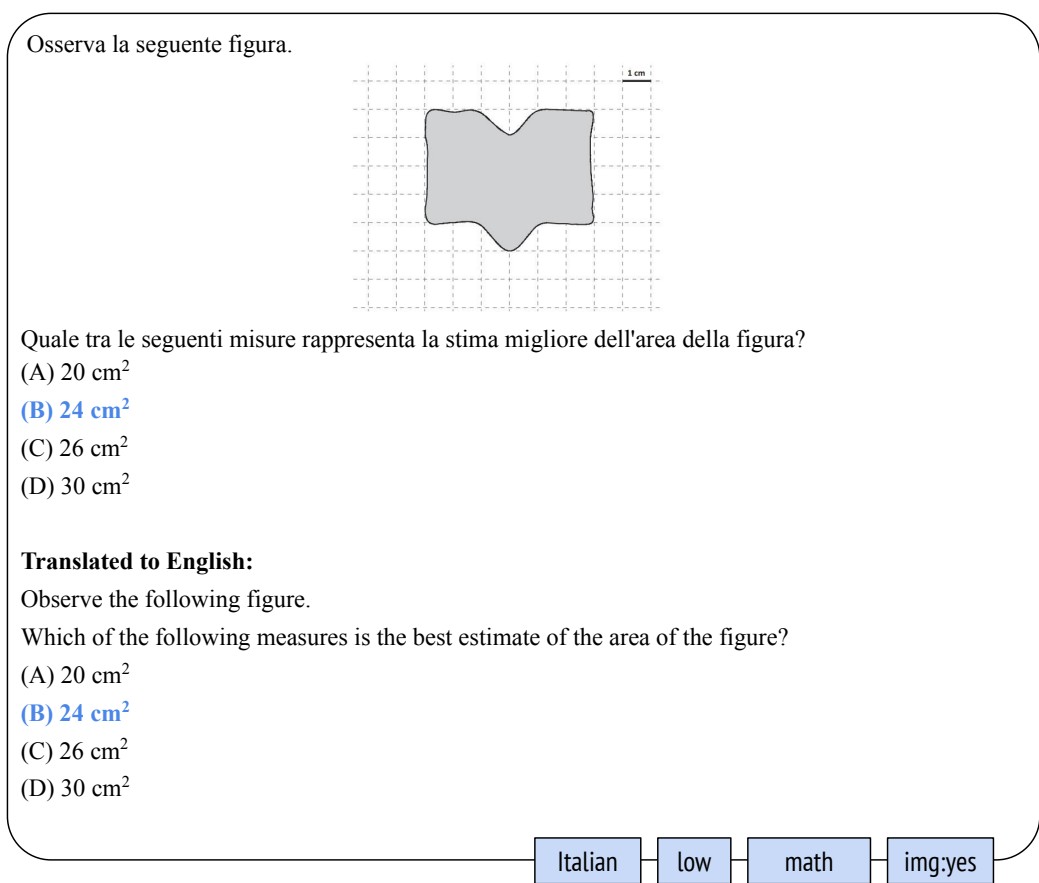

Quale tra le seguenti misure rappresenta la stima migliore dell'area della figura?

(A) 20 cm$^2$

**(B) 24 cm$^2$**

(C) 26 cm$^2$

(D) 30 cm$^2$

**Translated to English:**

Observe the following figure.

Which of the following measures is the best estimate of the area of the figure?

(A) 20 cm$^2$

**(B) 24 cm$^2$**

(C) 26 cm$^2$

(D) 30 cm$^2$

Italian — low — math — img:yes

Figure 10: An example question in Italian, with the correct answer in bold and blue.

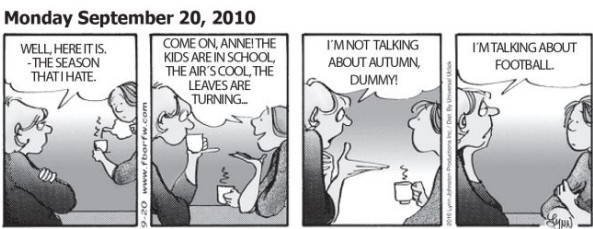

RIDGWAY, L. Disponível em: http://fborfw.com. Acesso em: 23 fev. 2012.

Na tira da série *For better or for worse*, a comunicação entre as personagens fica comprometida em um determinado momento porque

(A) as duas amigas divergem de opinião sobre futebol

(B) uma das amigas desconsidera as preferências da outra.

(C) uma das amigas ignora que o outono é temporada de futebol.

(D) uma das amigas desconhece a razão pela qual a outra a maltrata.

**(E) as duas amigas atribuem sentidos diferentes à palavra season.**

**Translated to English:**

In the series strip *For better or for worse*, the communication between the characters is compromised at a certain moment because

(A) the two friends have different opinions about soccer

(B) one of the friends disregards the preferences of the other

(C) one of the friends ignores that fall is football season

(D) one of the friends does not know why the other mistreats her

**(E) the two friends attribute different meanings to the word season**

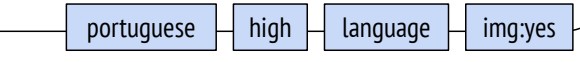

Figure 11: An example question in Portuguese, with the correct answer in bold and blue.

Trong hệ tiêu hóa của người, dưới tác động của enzim tiêu hóa, chất nào sau đây được biến đổi thành glixêrol và axit béo?

(A) Prôtêin

(B) Tinh bột

(C) Saccarôzơ

**(D) Lipit**

**Translated to English:**

In the human digestive system, as an effect of digestive enzymes, which of the following substances is broken down into glycerol and fatty acids?

(A) Protein

(B) Starch

(C) Sucrose

**(D) Lipids**

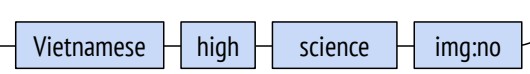

Figure 12: An example question in Vietnamese, with the correct answer in bold and blue.

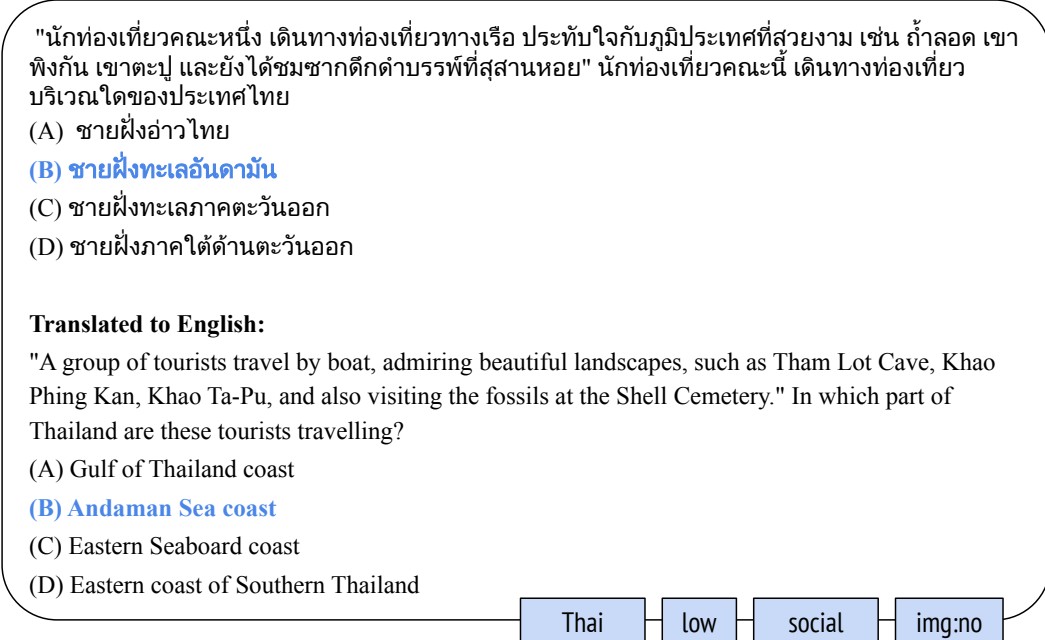

Figure 13: An example question in Thai, with the correct answer in bold and blue.

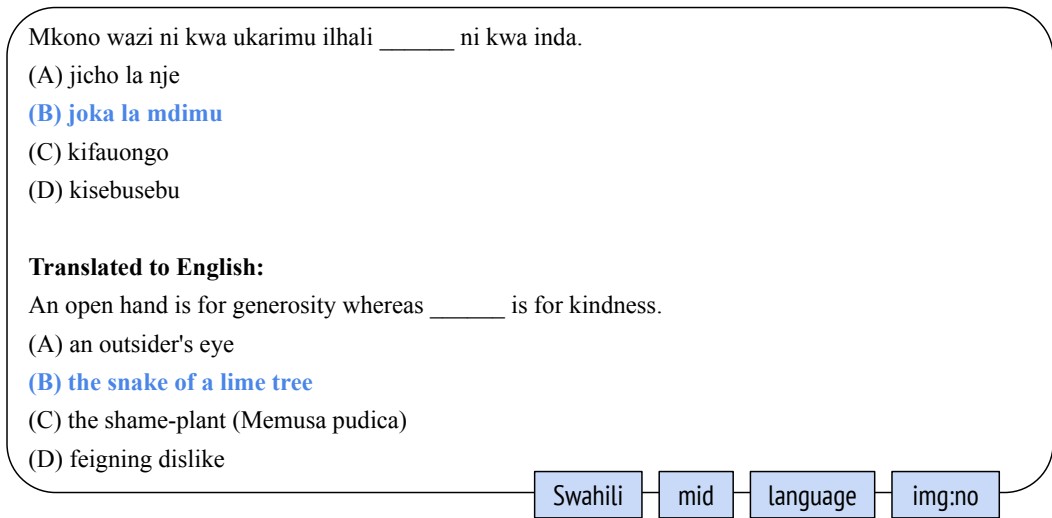

Figure 14: An example question in Swahili, with the correct answer in bold and blue.

Watter van die volgende beskryf Groot depressie van 1929 die beste?

(A) 'n Ekonomiese insinking

(B) Begin in die VSA

(C) die hele wêreld geraak

**(D) Al bogenoemde**

**Translated to English:**

Which of the following best describes the Great Depression of 1929?

(A) An economic slump

(B) Began in the USA

(C) Affected the whole world

**(D) All of the above**

Afrikaans — mid — social — img:no

Figure 15: An example question in Afrikaans, with the correct answer in bold and blue.

ꦏꦭ�023꧀ꦲꦶꦉꦭꦥꦶ Tulisan aksara jawa ing dhuwur yen ditulis aksara latin …

(A) ugur kuna sejati

(B) gugup rukun jati

(C) nunut kurun sejati

**(D) guyup rukun sejati**

**Translated to English:**

ꦏꦭ�023꧀ꦲꦶꦉꦭꦥꦶ The phrase in Javanese script above written in the Latin alphabet is …

(A) ugur kuna sejati

(B) gugup rukun jati

(C) nunut kurun sejati

**(D) guyup rukun sejati**

Javanese — low — language — img:no

Figure 16: An example question in Javanese, with the correct answer in bold and blue.

