# OpenReview forum: "M3Exam: A Multilingual, Multimodal, Multilevel Benchmark for Examining Large Language Models"
_NeurIPS.cc/2023/Track/Datasets_and_Benchmarks — NeurIPS 2023 Datasets and Benchmarks Poster_

### Official Review · Reviewer_uUHg · 2023-07-14
**Evaluating large language models using multilingual, multimodal, multilevel human exam questions**

**Rating:** 6
**Confidence:** 4
**Clarity:** Yes.

**Strengths:**

1. Evaluating the general intelligence of LLMs is an important research question and the paper is well-motivated by this research question;
2. The data collection process is well-documented;
3. The paper provides a fair analysis of seven state-of-the-art LLMs using the benchmark.

**Additional Feedback:**

See the Limitations section.

**Correctness:**

Generally yes; the reviewer has concerns about some claims comparing LLMs intelligence with human intelligence due to missing human performance.

**Documentation:**

Both data collection and code repo are well-documented.

**Ethics:**

No.

**Limitations:**

1. Human performance on the benchmark should be provided for direct comparison and more accurate evaluation of models (major weakness 1, 2);
2. It is also recommended to evaluate models on different subsets of datasets (for example, questions of different subjects, and multimodal questions grouped by different multimodal interactions that the model needs to capture to give correct answers) so the results can inform more about model behaviors beyond pure performance (major weakness 3).

**Opportunities For Improvement:**

Major weaknesses:
1. The proposed benchmark aims to evaluate the general intelligence of LLMs by human intelligence; however, human performance on the benchmark exam questions is not available for direct comparison;
2. It is mentioned that some questions are excluded from the original exams (line 130-132). Since in Table 2, model performance is compared with the conventional passing score, the reviewer believes it is necessary to either include human performance on the modified exam (major weakness 1), gauge the change in exam difficulty and adjust the standard for "passing" accordingly, or show that the change to exam questions does not affect the "passing" criteria;
3. Accuracy is the only evaluation metric, which is not very informative about model behaviors, especially in answering multimodal questions. For example, it does not show which aspect of processing and integrating multimodal signals the model fails on, such as model failure in aligning crucial information in the image with what is asked in the question (section 4.2). Moreover, questions are also grouped based on subjects, but this is not used in the following analysis or shown in the results.

Minor weaknesses:
1. More experiment details can be included (at least in the appendix). For example, data preprocessing should be clarified about multimodal models like BLIP that can only take in a single image as input and have a performance score on multi-image questions in Table 4.

**Relation To Prior Work:**

Yes.

**Summary And Contributions:**

This paper proposed a benchmark for evaluating the general intelligence of large language models. The benchmark consists of human exam questions in multiple languages (multilingual) and contains questions that must be solved by processing and integrating information from both text and image (multimodal). The exam questions are further structured into three levels corresponding to different learning stages in the development of human intelligence (multilevel). Seven state-of-the-art LLMs are evaluated on this benchmark and initial analysis shows the weakness of the models in processing questions in low-resource language and complex multimodal questions. The results also imply that the learning progress of LLMs differs from how humans learn.

---

> ### Author Response · Authors · 2023-08-18
> **Response to Reviewer uUHg**
>
> We would like to express our gratitude for your thorough review and thoughtful suggestions. We would like to respond to your comments as follows:
>
> **1. Lack of direct comparison with human performance**
>
> We appreciate your point here. However, the main objective of our work is not to compare the performance of models with humans directly. Rather, the random/passing performance serves more as an indicative benchmark. Given the large volume of questions included in this benchmark, it would be impractical to collect the statistics of the original exams (such as the distributions of the human exam-takers performance). A more equitable comparison could be to hire human annotators to answer (some sampled) questions, although the qualifications for these annotators would also need careful consideration. We will definitely think about this approach to establish more comprehensive results.
>
> In terms of the representative of the MCQs, we wish to clarify that no additional filtering was applied to these MCQs included in our dataset. Consequently, the difficulty level of these MCQs should be representative of the complete exam papers. This validates our use of conventional "passing" scores as effective indicators for human performance.
>
> **2. Suggestions on adding more analysis:**
>
> We are thankful for your suggestion of adding more dimensions of analysis beyond accuracy. In the paper, we primarily focused on three main aspects of this benchmark dataset, which is why we didn't include analysis of other aspects. However, the M3Exam dataset does provide rich meta-information (such as subject and level) that facilitates more detailed analysis for future work.
>
> Given the opportunity to add an additional page, we have included a section discussing results along different subject categories in the updated paper (see Sec 5 in the updated paper). We will consider deeper analysis to reveal the model's (success/failure) patterns in future work.
>
> Regarding the BLIP model, we would like to clarify that we only take the first image as the input, as it cannot process multiple images simultaneously. This was more of an investigation rather than a standard evaluation, and we have clarified this point in the revised paper. We appreciate you bringing this to our attention.

---

> > ### Comment · Reviewer_uUHg · 2023-08-22
> >
> > Thank you for your clarifications, new experiments, and discussion. Another partial cause for the prevalent failure on math questions may be the current challenge for LLM to process numbers and perform arithmetic operations. The authors' rebuttal has mostly addressed my concerns so I'm raising my rating to 6.

---

### Official Review · Reviewer_pzyt · 2023-07-22
**An informative multi-dimensional benchmark (multilingual, multi-level, and multi-modal) for evaluating LLMs**

**Rating:** 8
**Confidence:** 4

**Strengths:**

While an impressive amount of work focusing on real-world-exam-based evaluation of LLMs exists out there, M3Exam is, in my opinion, a much more comprehensive and meaningful effort in this direction. It is a benchmark that allows for multi-dimensional analysis of the reasoning and understanding capabilities of LLMs. Unlike most NLP benchmarks (e.g., XTREME-R), which arguably encompass many more languages, M3Exam is (1) "organic", in the sense that it is not obtained by translating English data but rather sourcing the questions from native countries/languages, (2) realistic, in that it encompasses real-world school exams (as opposed to tasks invented by NLP researchers as proxies for measuring model's reasoning and understanding abilities "in the wild"), and (3) multi-level, in that it divides questions into categories of different difficulty for the humans.

M3Exam is challenging benchmark, even for the most capable of the LLMs (e.g., GPT-4). As such, it could see wider adoption by the researcg community.


**Additional Feedback:**

One additional question: You say "BLOOM although a multilingual model gives unsatisfactory performance and is even worse than the random guess since it may generate invalid options". Why would you then not also evaluate (at least the open models, as you can obviously not do this for the closed commercial ones) in a constrained generation setup in which they are only allowed to generate tokens corresponding to answer options?


**Clarity:**

The paper is for the most part written clearly and it's easy to understand and follow. I did, however, identify a few ungrammatical sentences (examples below), so I'd advise the authors some more thorough (external) proof-reading.

"Gathering exam questions from varying educational levels is critical to assess and understand what level of intelligence that LLMs has been developed"
"The reason might be existing LLMs are already familiar with the question format of human exams"

**Correctness:**

The benchmark construction and evaluation seem to me correct and I could not identify any noteworthy shortcomings. Once could always discuss the (1) selection of languages for the benchmark and (2) LLMs for the comparative evaluation, but I believe both selections are sound. I would advocate for the inclusion of mT0 to the evaluated LLMs (as it was instruction fine-tuned multilingually and on a number of QA tasks, some I believe multi-choice), but this is just a minor comment/preference.

**Documentation:**

Adequate

**Ethics:**

I do not have any ethical concerns about this work.

**Limitations:**

My main cricitism of this work would be the lack of fine-tuning evaluations. The evaluation protocol, however, is predominantly limited to zero-shot inference (plus a bit of in-context few-shot inference). This measures the ability of LLMs to answer questions out of the box (i.e., as general-purpose generalizers) but not their ability to perform after being primed for these types of questions (i.e., fine-tuning for multi-choice). I definitely think that -- for the open models -- supervised fine-tuning on school exam questions would be warranted: given the immense sizes of these models, one would need to perform highly parameter-efficient fine-tuning (e.g., with low-rank adaptation, LoRA), but this is doable (e.g., with the most recent QLoRA, one can presumably do it even on regular consumer-grade cards). It would allow for a range of supervised transfer evaluations, across all three dimensions of the benchmark: cross-lingual transfer, cross-level transfer, and modality transfer (e.g., fine-tuning on text-only questions and evaluating on those including images and vice-versa).

**Opportunities For Improvement:**

I agree with the authors that obtaining questions from different geographies/languages naturally is better than translating them (from English or other major languages). However, there is a way to empirically test this: one could evaluate model's performance on both (1) organic questions in a language vs. (2) English translations to the language. While these would clearly have the confounding factors (foremost, that the two variants would not have the same questions) for comparing the performance of a single model, it would be interesting to see whether the ranking of the LLMs on the two would correlate. If yes, this would indicate that we could in principle use translated data -- which is much cheaper and easier to obrain -- for comparative evaluations of LLMs, even if the absolute performances of the models on translated data are optimistic compared to their performance on organic data.

**Relation To Prior Work:**

Prior work has been covered to a reasonable extent. The advantages of this benchmark compared to the existing ones are obvious (as discussed in the Strengths section).

**Summary And Contributions:**

In this work, the authors introduce a new benchmark for evaluation LLMs, one that consists of questions from real-world multi-choice school exams. The benchmark is multilingual -- with questions from exams from 9 different countries (and in 9 different languages), multimodal -- as some quarter of questions contain images, and multi-level -- with questions from different school levels: elementary, middle, and high-school. The authors then benchmark a wide range of state-of-the-art LLMs (some instruction-tuned, some not), showing that even the best among them (GPT-4) fail in many cases, especially for lower-resource languages of the benchmark (e.g., Javanese). Another interesting finding is that the LLMs, unlike humans, do not consistently perform better on "easier" questions from lower educational levels.

---

> ### Author Response · Authors · 2023-08-18
> **Response to Reviewer pzyt**
>
> We sincerely appreciate such a detailed and thoughtful comment, and your recognition of our work. We would like to address your points individually:
>
> **1. Translations of English questions to other languages:**
>
> It is an insightful suggestion of empirically testing the effectiveness of translated questions versus organically sourced questions in different languages. Indeed, it might serve as a proxy for the model's multilingual ability. However, as you pointed out, this approach might not fully encapsulate the local/native cultural nuances, which are crucial for a comprehensive evaluation. Given that our current focus is on creating a diverse and authentic dataset, we stick with real-world data in its original language. We do acknowledge the potential research opportunities by carefully comparing these two types of data in future work.
>
> **2. Experiments of fine-tuning setting:**
>
> Thanks for your thoughtful suggestions. The primary objective of the M3Exam benchmark is to evaluate the out-of-the-box performance of LLMs, which are expected to demonstrate a broad understanding of world knowledge and reasoning ability without task-specific training. Hence, the evaluation protocol primarily focuses on zero-shot inference and some in-context few-shot inference.
>
> We understand your point on the potential benefits of supervised fine-tuning, particularly for open models. We did conduct preliminary experiments with such settings, including continued training on additional unlabeled multilingual data with Llama 7/13B, using either Low-Rank Adaptation (LoRA) or full parameter fine-tuning. However, these efforts did not yield significant improvements, and the performance across different languages remained roughly the same. Our current observation is that the challenges presented in M3Exam may be difficult for even open tunable models of relatively smaller scales (such as 7b or 13B) to effectively handle. Nonetheless, we agree that this is an exciting direction to explore. We plan to train on larger models and more relevant data (like cleaned textbooks or exam-type MCQs) in our future work.
>
> **3. Writing:**
>
> Thank you for bringing these sentences to our attention. We will perform a more thorough proofreading process to ensure clarity and correctness in our writing.
>
> **4. Evaluation logic:**
>
> We appreciate your suggestion regarding the constrained generation setup. Our current evaluation logic is based on the text generated by the model. Although comparing the probability of different option letters could be an alternative approach, we believe it might provide overestimated results, as also discussed in the article [2]. Also, we think it's fairer to evaluate all models under the same evaluation logic, and thus take the generated text for computing the scores.
>
>
> [1] Llama 2: Open Foundation and Fine-Tuned Chat Models. CoRR abs/2307.09288
>
> [2] https://huggingface.co/blog/evaluating-mmlu-leaderboard

---

### Official Review · Reviewer_RvNv · 2023-07-22
**An interesting work**

**Rating:** 7
**Confidence:** 4
**Clarity:** Yes

**Strengths:**

1 This work presents a human exam questions benchmark, with multilingual, multimodal, and multilevel questions .
2 This work presents some interesting findings.
3 Both dataset and codes are open source.

**Additional Feedback:**

No.

**Correctness:**

The dataset was constructed in a sound way.
The evaluation methods and experiment design are appropriate and performed correctly.

**Documentation:**

Done.

**Ethics:**

None.

**Limitations:**

1 The results in Table 3 seem not good. What are the reasons (data or model problems)?
2 Limited novelty.

**Opportunities For Improvement:**

1 Some statements are lack of detailed explaination. For example, the statement in Lines 48-50, "placeholder" in line 144.
2 Figure 5 in line 243 is wrong.
3 Give a deep analysis about the experimental results.

**Relation To Prior Work:**

Clearly discussed.

**Summary And Contributions:**

This paper proposes a human exam questions benchmark  for evaluating LLMs in a multilingual, multimodal, and multilevel context.

---

> ### Author Response · Authors · 2023-08-18
> **Response to Reviewer RvNv**
>
> Thank you for your feedback on several parts of our paper. We have clarified them in the updated paper accordingly. In specific:
>
> 1. Lines 48~50: It is referring to potential issues with multilingual benchmarks for traditional NLP tasks that have been created by translating original English datasets. While translation allows these benchmarks to be used for testing models across different languages, it can inadvertently introduce an English-centric bias. While the words can be translated, the underlying concepts or ideas they represent in English may not exist or may be understood or interpreted differently in other languages. For example, a history question might ask "What is the King of England in 1900?". Though this English question can be translated into other languages, it is less likely to be a question that a native speaker of those languages might be interested in or use in their real-world applications.
> 2. Line 144: It means the annotators will use a special mark at the place of an image. For example, \texttt{(image)[image-x.jpg]} will appear in the place of an image in the transformed text, and the corresponding image will be clipped and saved with the same name (i.e., image-x.jpg).
> 3. Line 243: Figure 5 is given in the Appendix, due to the separation requirement of the submission, they are split into two PDFs. We have made this clear in the updated paper.
> 4. More detailed result analysis in Table 3: Table 3 shows results under different settings for a more detailed analysis. We explained and analyzed the performance of these settings in Lines 237~266.

---

### Official Review · Reviewer_JYpB · 2023-07-24
**Valuable Resource. Experiments could be better.**

**Rating:** 7
**Confidence:** 4
**Clarity:** Yes.

**Strengths:**

1. The dataset proposed in this paper covers nine different languages, many of which are low-resource languages. This kind of benchmark can help promote the development of more inclusive LLMs. Similarly, multimodality is a key issue for LLMs, and new datasets in this field are valuable. I believe the dataset proposed in this paper will be helpful for future research work.

2. The writing quality of this paper is very good, with the content being clear and easy to follow.

3. The viewpoint mentioned in Section 4.3, that the learning difficulties of LLMs may not align with those of humans, is a common hypothesis. Discovering empirical evidence to support this hypothesis is valuable for our future understanding of the mechanisms of LLMs.

**Additional Feedback:**

None.

**Correctness:**

Generally good, but there is room for improvement, see Opportunities For Improvement #3 & #4.


**Documentation:**

Yes.

**Ethics:**

The paper uses a lot of examination papers as data sources, and I'm not sure whether this infringes on copyright. The paper does not explicitly discuss this.

**Limitations:**

This paper does not discuss potential negative societal impacts, but I believe the potential negative impacts that this work might cause are limited.


**Opportunities For Improvement:**

1. The results reported by the benchmark proposed in this paper are rather coarse-grained, only providing evaluation results according to language, presence of images, educational periods, etc. It would be more helpful for the development of LLMs if more targeted diagnostic results could be obtained based on finer-grained meta-information such as disciplines, relationships between knowledge, etc.

2. The highest difficulty level of the questions in this dataset only goes up to high school, which presents limited challenges compared to similar works like MMLU, AGIEval, and C-Eval.

3. The range of LLMs covered by the experiments in this paper is relatively narrow, which limits the value of the results reported in this paper as references for users choosing LLMs.

4. Some of the interesting experimental results in this paper could benefit from further detailed analysis. For example, in Table 4, Flan-T5 achieved non-trivial performance without image information, which might affect our judgment of data quality. In the results of Figure 4, although the performance at the middle level is higher than that at the low level, the margin is quite small in most languages. I believe it is necessary to analyze whether there is a difference in the data quality between the two levels to draw valid conclusions.

**Relation To Prior Work:**

Yes.

**Summary And Contributions:**

This paper proposes M3Exam, a benchmark for evaluating large language models (LLMs) with three features: multilingual, multimodal, and multilevel. It covers 12,317 questions in 9 languages, among which 2,816 questions require understanding images. The questions are divided into three levels by education periods: primary, middle, and high school. This paper also conducts experiments with a few recent LLMs and multi-modal models.

---

> ### Author Response · Authors · 2023-08-18
> **Response to Reviewer JYpB**
>
> We thank you for your acknowledgment of all three aspects of our benchmark! Detailed responses to your comments are given below:
>
> **1. The suggestion of reporting results based on finer-grained meta-information**
>
> We acknowledge your suggestion for a more granular analysis of the evaluation results, which could indeed provide more detailed insights. In the scope of our paper, we decided to focus primarily on the multilingual, multimodal, and multilevel aspects due to space constraints. However, the M3Exam dataset is designed to support a more comprehensive examination. It provides rich meta-information for all questions, including subject, subject category, and more, allowing for additional targeted diagnostic evaluations.
>
> Following your suggestion, we add a section to analyze the performance variance across different subject categories (see Sec 5 in the updated paper). A clear pattern is that almost all languages fall short in the math subject, either it is a high-resource language such as English or a low-resource language such as Thai, possibly due to the difficulty of reasoning-type questions in math subjects. We agree detailed analysis might provide more interesting findings and we look forward to subsequent research leveraging the rich meta-information in our benchmark for further exploration and analysis.
>
> **2. Limitations on the question difficulty**
>
> Your observation about the highest difficulty level in our M3Exam dataset is accurate. However, as discussed at the end of Section 4.3, we find that the development of model intelligence does not necessarily align with humans. Our results show that even advanced models like GPT-4 struggle with questions from the primary school level, which suggests that higher complexity does not always equate to a more effective evaluation of model intelligence. We believe that the current range of questions already poses a great challenge for various LLMs.
>
> **3. Selection of evaluated models**
>
> We appreciate your feedback regarding the range of LLMs evaluated in our paper. Our selection was based on the unique requirements of M3Exam that demand multilingual & multimodal capabilities. Many models were naturally excluded due to these specifications. We have indeed made an effort to include a diverse range of top-performing models, both open-source and closed-source, for the multilingual evaluation. When it comes to multimodal abilities, we evaluated top-performing models like BLIP-2 and InstructBLIP (which are quite recent). We plan to expand our evaluation as new models become available. For instance, should GPT-4 introduce the multimodal input API, we would certainly consider it for evaluation.
>
> **4. Suggestions on adding more analysis**
>
> Thank you for your insightful suggestions. We agree that further detailed analysis of these experimental results could provide additional valuable insights. We would also like to clarify that we have very strict data processing where multiple rounds of quality checks have been conducted from time to time to ensure the final data quality. Therefore, the results indicate the different weaknesses or issues of current LLMs rather than issues with the data quality of M3Exam. For example, the performance gap between low and mid levels is indeed marginal, that's why we give the observation that "there is no clear decreasing trend as the level increases" in the paper. We also added more discussions according to your suggestions. We believe these discussions provide a more nuanced understanding of our results and we appreciate your interest in these important details.
>
> **5. Concerns on potential ethical issues**
>
> Thank you for raising this point regarding copyright considerations. We have carefully explained such an issue in detail in the response to the ethical reviewer.

---

> > ### Comment · Reviewer_JYpB · 2023-08-29
> >
> > Thanks for the careful response. I will maintain my positive score and wish the authors good luck.

---

### Official Review · Reviewer_grkh · 2023-07-25
**Nice benchmark for LLM evaluation**

**Rating:** 7
**Confidence:** 4
**Correctness:** The design, collection and evaluation…
**Clarity:** Yes, the paper is well written.

**Strengths:**

The proposed benchmark, M3Exam, hits the middle of multilingual, multimodal and multilevel evaluation, which should be valuable to the research community and could facilitate the research on LLMs.

**Additional Feedback:**

* Did you check whether the used exams are, in any chance, leaked to public web data (e.g. common crawl)?
* While GPT4 and ChatGPT achieve good performance on M3Exam, whether we could call it zero-shot results is questionable. We have no idea what finetuning data GPT models have used.

**Documentation:**

The paper has given proper details on data documentation.

**Ethics:**

No.

**Limitations:**

M3Exam only covers multiple-choice question-answering tasks, which can't fully examine LLM's generation capability.

**Opportunities For Improvement:**

The multimodal evaluation is concerned: text-only models largely outperform vision-text models. It would be great to add ChatGPT and GPT-4 results to Table 4 to better understand the upper bound performance achieved by text-only models.

**Relation To Prior Work:**

Related works are properly discussed.

**Summary And Contributions:**

Large language model (LLM) has shown impressive performance across a wide range of tasks and exhibited ability beyond language understanding. How to properly evaluate it has never been so crucial ever before. This paper follows the idea of leveraging human exams to test the capability of LLMs and introduces M3Exam, featuring three aspects: multilingual, multimodal, and multilevel. Results of several top-performed LLMs on M3Exam reveal that 1) existing LLMs' multilingual capability is limited, still falling short on non-English languages, and 2) they still struggle with understanding complex image inputs.

---

> ### Author Response · Authors · 2023-08-18
> **Response to Reviewer grkh**
>
> Thanks for your constructive feedback! Our detailed responses to your comments are as follows:
>
> **1. Add stronger models for the multimodal evaluation**
>
> Initially, we only leverage the Flan-T5 model as the text-only baseline model for the multimodal questions since it has a similar model size as other multimodal models and serves as the text backbone of many of them, making the comparison more straightforward. And we agree that the results are indeed "concerned" -- multimodal models do not show clear improvements, indicating there is a large room for current multimodal models to improve their understanding ability on images requiring complex reasoning such as those in M3Exam.
>
> We appreciate your suggestion to incorporate results from more powerful text models to better understand the upper bound of their performance. In response, we have included the results obtained using ChatGPT in the updated paper (see Table 4). As anticipated, ChatGPT does outperform Flan-T5, which suggests that text-only models can indeed extract useful information to question text only. However, a significant performance gap remains evident when comparing Tables 2 and 3 (which are supposed to be comparable for the same language since they come from the same exam papers). This shows that many questions must require image-based information, thereby posing significant challenges for current multimodal models.
>
>
> **2. Feedback on potential data contamination**
>
> To construct our benchmark, we found that most original exam papers have separate files for the question and answer (which is similar to the findings in MMLU [1]). Thus the chance of data contamination is very low. We will conduct a more systematic investigation in future work.
>
>
> **3. Feedback on the name of "zero-shot results"**
>
> We appreciate your comment about the ambiguity in defining "zero-shot" results. In our paper, we consider GPT-4 and ChatGPT's performance as "zero-shot" in the context that they were not specifically fine-tuned on the M3Exam benchmark before evaluation. We will make this more straightforward in the paper.
>
>
> [1] Measuring Massive Multitask Language Understanding. ICLR 2021

---

### Decision · Program_Chairs · 2023-09-22

**Decision:**

Accept (Poster)

**Comment:**

This paper proposes M3Exam, a benchmark for evaluating large language models (LLMs) with three features: multilingual, multimodal, and multilevel. It covers 12,317 questions in 9 languages, among which 2,816 questions require understanding images. The questions are divided into three levels by education periods: primary, middle, and high school. This paper also conducts experiments with a few recent LLMs and multi-modal models.
Pros:
1)The benchmark allows for multi-dimensional analysis of the reasoning and understanding capabilities of LLMs, which is from organic, realistic and multi-level school exams.
2)The benchmark covers nine different languages, many of which are low-resource languages.
3)The benchmark is challenging, even for the most capable of the LLMs (e.g., GPT-4). As such, it could see wider adoption by the research community.
4)The writing quality of this paper is very good, with the content being clear and easy to follow.
Cons:
1)The benchmark only covers multiple-choice question-answering tasks, which can't fully examine LLM's generation capability.
2)The evaluation results reported by the benchmark proposed in this paper are rather coarse-grained. It is proposed to provide diagnostic results obtained based on finer-grained meta-information such as disciplines, relationships between knowledge, etc.
3)The work is lack of fine-tuning evaluations. It would allow for a range of supervised transfer evaluations, across all three dimensions of the benchmark: cross-lingual transfer, cross-level transfer, and modality transfer.

In summary, all reviewers agreed that this is very solid submission and authors also handled concerns from reviewers during discussion period. I recommend acceptance.